# Engineering n-Type and p-Type BiOI Nanosheets: Influence of Mannitol on Semiconductor Behavior and Photocatalytic Activity

**DOI:** 10.3390/nano14242048

**Published:** 2024-12-21

**Authors:** Shuo Yang, Wenhui Li, Kaiyue Li, Ping Huang, Yuquan Zhuo, Keyan Liu, Ziwen Yang, Donglai Han

**Affiliations:** 1School of Materials Science and Engineering, Changchun University, Changchun 130022, China; 240501286@mails.ccu.edu.cn (W.L.); 220502210@mails.ccu.edu.cn (K.L.); 230502246@mails.ccu.edu.cn (P.H.); 240501261@mails.ccu.edu.cn (Y.Z.); 2Laboratory of Materials Design and Quantum Simulation College of Science, Changchun University, Changchun 130022, China; 3School of Materials Science and Engineering, Changchun University of Science and Technology, Changchun 130022, China; 2022101221@mails.cust.edu.cn (K.L.); 2024101323@mails.cust.edu.cn (Z.Y.)

**Keywords:** BiOI nanosheets, photocatalysts, mannitol, n-type, p-type

## Abstract

Photocatalytic technology holds significant promise for sustainable development and environmental protection due to its ability to utilize renewable energy sources and degrade pollutants efficiently. In this study, BiOI nanosheets (NSs) were synthesized using a simple water bath method with varying amounts of mannitol and reaction temperatures to investigate their structural, morphological, photoelectronic, and photocatalytic properties. Notably, the introduction of mannitol played a critical role in inducing a transition in BiOI from an n-type to a p-type semiconductor, as evidenced by Mott–Schottky (M-S) and band structure analyses. This transformation enhanced the density of holes (h^+^) as primary charge carriers and resulted in the most negative conduction band (CB) position (−0.822 V vs. NHE), which facilitated the generation of superoxide radicals (·O^2−^) and enhanced photocatalytic activity. Among the samples, the BiOI-0.25-60 NSs (synthesized with 0.25 g of mannitol at 60 °C) exhibited the highest performance, characterized by the largest specific surface area (24.46 m^2^/g), optimal band gap energy (2.28 eV), and efficient photogenerated charge separation. Photocatalytic experiments demonstrated that BiOI-0.25-60 NSs achieved superior methylene blue (MB) degradation efficiency of 96.5% under simulated sunlight, 1.14 times higher than BiOI-0-70 NSs. Additionally, BiOI-0.25-60 NSs effectively degraded tetracycline (TC), 2,4-dichlorophenol (2,4-D), and rhodamine B (Rh B). Key factors such as photocatalyst concentration, MB concentration, and solution pH were analyzed, and the BiOI-0.25-60 NSs demonstrated excellent recyclability, retaining over 94.3% of their activity after three cycles. Scavenger tests further identified ·O^2−^ and h^+^ as the dominant active species driving the photocatalytic process. In this study, the pivotal role of mannitol in modulating the semiconductor characteristics of BiOI nanomaterials is underscored, particularly in promoting the n-type to p-type transition and enhancing photocatalytic efficiency. These findings provide a valuable strategy for designing high-performance p-type photocatalysts for environmental remediation applications.

## 1. Introduction

The rapid industrial growth in recent decades has significantly exacerbated water contamination, posing severe environmental and societal challenges [1,2]. Among the various pollutants affecting water bodies, dye pollution is particularly concerning. Methylene blue (MB), a widely used cationic dye, is one of the major contributors to this issue [3,4,5]. Frequently employed in industrial processes, MB residues in water present substantial threats to both ecological systems and human health [6,7]. Its presence disrupts aquatic ecosystems, jeopardizes public health, impedes economic development, and undermines overall social well-being [8,9,10]. Given these far-reaching impacts, the effective elimination of MB from contaminated water has become an urgent priority [11,12]. To tackle this issue, researchers have explored diverse remediation strategies, including biological, chemical, physical, and ultrasonic techniques [13,14,15]. However, these approaches often suffer from inherent limitations, such as complex pretreatment processes, high operational costs, and limited scalability [16,17]. Therefore, there is an urgent need for a remediation method that is not only cost-effective and environmentally sustainable but also capable of efficiently degrading dye contaminants [18,19,20].

Among the various approaches to addressing energy challenges and water pollution, photocatalytic technology has attracted significant attention for its excellent performance in efficiently degrading various organic pollutants in water and facilitating chemical transformations [21,22]. Photocatalytic technology utilizes photogenerated electrons and holes generated by semiconductor materials under light irradiation, driving redox reactions to effectively degrade organic pollutants or achieve other chemical transformations [23,24,25]. The efficiency of this process is critical, as it dictates the efficiency of photocatalytic degradation process [26,27]. Among the numerous photocatalysts, Bi-based materials stand out due to their unique [Bi_2_O_2_] layered structure [28,29,30]. This special structure facilitates the formation of an internal electric field between the layers, which enhances the separation of photogenerated electron-hole (e^−^ − h^+^) pairs and improves photocatalytic efficiency [31,32,33]. Among Bi-based photocatalysts, BiOI is a particularly promising candidate due to its relatively narrow direct bandgap, which enables the significant absorption of visible light compared to other Bi-based materials [34,35]. This narrow bandgap enables BiOI to exhibit significant absorption of visible light [36,37]. Additionally, the properties and composition of BiOI can be effectively tuned through synthesis methods and processing conditions, further enhancing its performance [38,39,40].

Currently, various methods have been employed for the preparation of BiOI semiconductor materials, including coprecipitation [41], microemulsion [42], sol–gel [43], hydrothermal synthesis [44], and water bath [45], among others. For instance, Mera A. C. et al. [46] synthesized BiOI microspheres via the solvothermal method and achieved 60% gallic acid removal under UV-A irradiation within 30 min using ionic liquids. Chang X. et al. [47] synthesized a BiOI nanosheet using co-precipitation, which reported the efficient photocatalytic elimination of sodium pentachlorophenol (PCP-Na) under artificial sunlight illumination, achieving a degradation efficiency of 90.3% within one hour. Ren K. et al. [48] prepared flower-like BiOI microspheres utilizing a simple precipitation method in a low-temperature (70 °C) mixed water–ethanol solution via the aid of PVP and citric acid (CA), although the maximum photodegradation efficiency was limited to 40%. Among all these methods, the water bath method has been recognized as the simplest, most cost-effective, and least toxic approach for synthesizing BiOI materials [45].

The enrichment of active sites plays a pivotal role in enhancing the efficient separation of photogenerated electron-hole pairs, a process that directly dictates photocatalytic activity. For example, Hu H. et al. [49] synthesized activated carbon-supported molybdenum oxide (Mo/AC) catalysts that demonstrated superior stability and catalytic performance in the epimerization of glucose to mannose, achieving a selectivity of 94.5%. Similarly, Chen C. et al. [50] utilized mannitol-modified Cu/SiO_2_ catalysts to optimize the distribution of copper species, thereby enhancing the synergistic effect of Cu^+^ species, which led to high activity and stability in the selective hydrogenation of DMO to EG. Zheng Y. et al. [51] investigated the doping of D-sorbitol into PEDOT:PSS and found that a 4 wt.% addition significantly boosted the solar cell’s short-circuit current (from 5.81 to 15.91 mA/cm^2^) and power conversion efficiency (from 2.50% to 3.62%). Inspired by the role of active agents in enhancing catalytic properties, we employed mannitol as the active agent in our synthesized catalysts. Mannitol was selected due to its non-hygroscopic nature, ease of separation from liquid-phase products, and excellent recyclability. Using a parameter-specific water bath method, we achieved a higher concentration of active sites, which significantly enhanced the surface activity of the catalyst. This, in turn, facilitated the efficient separation and transfer of photogenerated electrons and holes to the photocatalyst surface, promoting the overall photocatalytic performance. Although extensive research has been conducted on BiOI as a photocatalyst, its photocatalytic degradation efficiency remains suboptimal. Furthermore, comprehensive and systematic investigations into the structure, morphology, optoelectronic properties, and catalytic mechanisms of BiOI are notably scarce. Thus, developing a simple and efficient approach to synthesize high-performance BiOI photocatalysts, alongside in-depth and systematic studies of their structural, morphological, and optoelectronic characteristics, as well as their catalytic mechanisms, is of critical importance.

In the present work, we utilized a simple and cost-effective water bath method to synthesize a series of BiOI NS photocatalysts for the degradation of MB, Rh B, TC, and 2,4-D in water. By precisely controlling the amount of mannitol added, we successfully modulated the p-type and n-type characteristics of BiOI NSs and further optimized their photocatalytic performance by adjusting the synthesis temperature. Notably, the incorporation of mannitol and the precise regulation of its dosage significantly enhanced the photocatalytic activity of BiOI semiconductors. This enhancement is likely due to its effects on crystal growth, lattice defect modulation, the formation of a surface passivation layer, and alterations in the chemical environment and interfacial properties [52,53,54]. In this study, the effects were systematically examined of mannitol dosage and synthesis temperature on the structure, morphology, optoelectronic properties, and photocatalytic performance of BiOI NSs, along with its photocatalytic mechanism. Special attention was given to the role of mannitol in driving the transition between p-type and n-type semiconductor characteristics, as well as its influence on the bandgap position and overall photocatalytic efficiency of BiOI NSs.

## 2. Materials and Methods

### 2.1. Chemicals and Reagents

The materials used in these experiments included bismuth nitrate pentahydrate [Bi(NO_3_)_3_·5H_2_O, AR, 99%], purchased from Shanghai Macklin Biochemical Technology Co., Ltd., Shanghai, China, while mannitol (C_6_H_14_O_6_, AR, 98%), methylene blue (MB, AR), potassium iodide (KI, AR, ≥99.5%), Rhodamine B (Rh B, AR), 2.4-Dichlorophenol (C_6_H_4_C_l2_O, AR), tetracycline (TC, AR), sodium hydroxide (NaOH, 99%), hydrochloric acid (HCl, 99%), anhydrous ethanol (CH_3_CH_2_OH, AR), 4-Hydroxy-TEMPO (TEMPO, AR, 98%), isopropyl alcohol (IPA, AR, ≥99.5%), and ethylene-diaminetetraacetic acid disodium salt (EDTA-2Na, AR, 0.1 M) were obtained from Shanghai Aladdin Biochemical Technology Co., Ltd., Shanghai, China. All reagents were used without any further modification or purification prior to the experiments.

### 2.2. BiOI NS Preparation

BiOI NSs were synthesized using a controlled and reproducible method. Initially, 2.425 g of Bi (NO_3_)_3_·5H_2_O was weighed, thoroughly ground, and dispersed in 60 mL of deionized water under magnetic stirring for 1 h. Subsequently, varying amounts of C_6_H_14_O_6_ (mannitol) were added to the solution in the following quantities: 0 g, 0.1 g, 0.15 g, 0.2 g, 0.25 g, 0.3 g, and 0.35 g. In parallel, 0.83 g of KI was dissolved in 40 mL of deionized water and magnetically stirred for 1 h. The prepared KI solution was then gradually introduced into the Bi(NO_3_)_3_·5H_2_O solution along the inner wall of the beaker while maintaining continuous stirring at a constant temperature of 70 °C for 3 h. The resulting mixture was repeatedly washed with anhydrous ethanol and deionized water to remove impurities. The final product was vacuum-dried at 60 °C for 12 h, yielding BiOI NSs. These samples were labeled as BiOI-0-70, BiOI-0.1-70, BiOI-0.15-70, BiOI-0.2-70, BiOI-0.25-70, BiOI-0.3-70, and BiOI-0.35-70, based on the amount of mannitol used. To further evaluate the effect of synthesis temperature, the same procedure was performed at constant temperatures of 40 °C, 50 °C, 60 °C, and 80 °C, using 0.25 g of mannitol. The resulting samples were designated as BiOI-0.25-40, BiOI-0.25-50, BiOI-0.25-60, BiOI-0.25-80, BiOI-0.25-50, BiOI-0.25-60, and BiOI-0.25-80, respectively.

### 2.3. Characterization

The physical phase analysis was conducted using X-ray diffraction (XRD) on a Bruker D2 diffractometer, manufactured by Bruker AXS GmbH, Karlsruhe, Germany. Fourier transform infrared (FT-IR) spectroscopy was captured on an IRTracer-100 instrument, manufactured by Shimadzu Scientific Instruments (SSI), Columbia, MD, USA, to identify functional group species. Chemical composition and binding energies were determined using X-ray photoelectron spectroscopy (XPS) with a Thermo Escalab 250Xi instrument, which is manufactured by Thermo Fisher Scientific Inc., a company based in Waltham, Massachusetts, USA. The morphology and microstructure of the synthesized samples were examined using scanning electron microscopy (SEM) with a JSM-7500F instrument from Nippon Electronics Corporation in Tokyo, Japan, and transmission electron microscopy (TEM) with a Tecai F20 instrument from FEI USA in Hillsboro, USA. UV-Vis diffuse reflectance spectra (DRS) were recorded using a Shimadzu UV-3600 plus UV-Vis spectrophotometer, manufactured by Shimadzu Scientific Instruments (SSI), headquartered in Columbia, MD, USA. Photoluminescence spectra (PL) were analyzed using a Hitachi F-4700 spectrometer, manufactured by Hitachi High-Technologies Corporation, Tokyo, Japan. Impedance spectra (EIS) and photocurrent measurements were conducted utilizing a CHI 66E electrochemical workstation, supplied by Shanghai Chenhua Instrument Co., Ltd., based in Shanghai, China. The specific surface area was analyzed using the Brunauer-Emmett-Teller (BET) method with a JW-BK200C instrument from Beijing JWGB Sci & Tech Co., Ltd., located in Beijing, China.

### 2.4. Photocatalytic Degradation Experiment

The photocatalytic activity of the samples was evaluated by degrading an MB aqueous solution under simulated sunlight provided by a 500W xenon lamp (CEL-S5500/350). A total of 10 mg of photocatalyst was added to a 10 mg/L MB solution. To eliminate the influence of photocatalyst adsorption and dye sensitization effects on photocatalytic degradation efficiency, the mixture was magnetically stirred in the dark for 40 min to reach adsorption–desorption equilibrium before starting the photocatalytic test. The MB solution to be degraded was placed 15 cm from the lamp, after which illumination commenced. At 10 min intervals, the upper suspension was extracted and centrifuged. The absorption peak intensity of the methylene blue solution at 662 nm was monitored with a UV–visible spectrophotometer. The photocatalytic degradation rate was calculated by the following equation:*DR% = [1 − (C/C_0_)] × 100%*(1)
where *C* and *C*_0_ stand for the concentrations of the MB solution at the beginning and time *t*, respectively. The following equation pseudo-first-order dynamical models were used to fit the MB degradation data:*−ln(C/C_0_) = kt*(2)
The apparent rate constant (min^−1^) is denoted by *k*. Following each cycle, the samples’ photocatalytic activity was assessed, the photocatalytic time as well as the photocatalytic conditions were not changed, and the catalysts were washed with anhydrous ethanol and sonicated several times after each experiment. Repeated five times, this process confirms the reproducibility of the photocatalytic results.

## 3. Results and Discussion

### 3.1. Phase Structures and Morphologies

Figure 1a,b display the XRD patterns of BiOI NSs synthesized with varying mannitol concentrations and reaction temperatures. As shown in Figure 1a,b, the diffraction peaks located at 9.66°, 19.36°, 24.29°, 29.65°, 31.66°, 37.39°, 39.37°, 45.38°, 51.35°, and 55.15° corresponded to the diffraction peaks at the (001), (002), (101), (102), (110), (112), (004), (200), (114), and (212) crystal planes of the tetragonal BiOI, aligning with the standard JCPDS card number 10-0445 [55]. Comparing the XRD patterns of samples prepared with varying amounts of mannitol [BiOI-(0~0.35)-70 NSs] and those synthesized with 0.25 g of mannitol under different reaction temperatures [BiOI-0.25-(40~80) NSs] revealed no additional impurity peaks or significant shifts in diffraction peak positions. This confirms the lattice structural stability of the BiOI NSs synthesized by this method [56,57]. Figure 1c shows the FT-IR spectra of BiOI-0.25-60 and BiOI-0-70 NSs. The peaks observed at 667 cm^−1^ and 920 cm^−1^ are attributed to the stretching vibration of the Bi-O bond [58]. The peak around 1060 cm^−1^ and the broad absorption band at approximately 2930 cm^−1^ are due to the vibration of the C-H bond [59]. The peak at 1381 cm^−1^ is possibly attributed to the stretching vibration of the I-O bond [60]. The absorption peaks at 1630 cm^−1^ and 3451 cm^−1^ correspond to water adsorbed on the surface of BiOI [59]. Additionally, the peak at 2342 cm^−1^ is caused by the C=O bond in the CO_2_ molecule [61]. These findings further confirm the successful synthesis of the BiOI NSs.

To gain deeper insight into the elemental composition and chemical states of the synthesized nanocomposites, XPS analysis was conducted. Figure 1d–h display the XPS survey spectra of BiOI-0.25-60, along with the high-resolution XPS spectra for the Bi, O, I, and C elements. Figure 1d indicates that the surface of BiOI-0.25-60 contains the Bi, O, I, and C elements. Figure 1 e illustrates the Bi 4f spectrum, which features two peaks at 158.7 eV and 164.0 eV, corresponding to the Bi^3+^ 4f_7/2_ and 4f_5/2_ states in BiOI [62]. In Figure 1f, the I 3d spectrum displays two peaks at 618.6 eV and 630.2 eV, which are assigned to the I 3d_5/2_ and I 3d_3/2_ orbitals, respectively [63]. Figure 1g displays the O 1s spectrum, with two peaks located at 530.2 and 532.7 eV, attributed to Bi-O bonding and oxygen adsorbed in oxygen vacancies, respectively [64]. As shown in Figure 1h, the C 1s peak centered at 285 eV is primarily attributed to the CHx functional group in the mannitol molecule absorbed on BiOI-0.25-60 [65]. Collectively, these XPS results further confirm the successful synthesis of BiOI NSs.

The pore properties and specific surface areas of BiOI-0.25-60 and BiOI-0-70 NSs were analyzed through N_2_ adsorption–desorption tests. The N_2_ adsorption–desorption isotherms of BiOI-0.25-60 and BiOI-0-70 NSs are shown in Figure 2a,b. Both the BiOI-0.25-60 and BiOI-0-70 NSs exhibit type IV isotherms with H3 hysteresis loops. The BET specific surface area of BiOI-0-70 NSs is 7.12 m^2^/g, while that of BiOI-0.25-60 NSs is 24.46 m^2^/g. This increase in surface area enhances the photocatalyst’s adsorption capacity, providing additional active sites for photocatalytic reactions, thereby improving photodegradation efficiency. As depicted in Figure 2c,d, the average pore diameters of BiOI-0.25-60 and BiOI-0-70 NSs are 23.52 nm and 12.82 nm, respectively, indicating that both materials possess a mesoporous structure. Compared with BiOI-0-70 NSs, the BiOI-0.25-60 NSs have larger pore sizes, which can provide more efficient transport paths for carriers to enter the interior of the samples, helping to shorten carrier diffusion paths and enhance photocatalytic activity [66,67,68]. The large specific surface area and pore size of BiOI-0.25-60 NSs may result from the role of mannitol as a template or pore-forming agent during the reaction, influencing the crystal growth process and leading to BiOI with a higher surface area and a more porous structure.

Figure 3a–g show the SEM images of BiOI NSs synthesized at 70 °C with varying amounts of mannitol (0 g, 0.1 g, 0.15 g, 0.2 g, 0.25 g, 0.3 g, 0.35 g). Figure 3h–l present SEM images of BiOI NSs prepared with 0.25 g of mannitol at different temperatures (40 °C, 50 °C, 60 °C, 80 °C). Figure 3m–p display the EDS mapping of BiOI-0.25-60 NSs. As shown in Figure 3a–l, all samples exhibit irregular nanosheet morphology with a size distribution ranging from 40 to 350 nm. The average size of the BiOI NSs significantly decreases with increasing mannitol dosage (Figure 3a–g), whereas the temperature variation has minimal impact on the morphology of the BiOI NSs (Figure 3h–l). As indicated in Figure 3a m–p, the Bi, O, and I elements are uniformly distributed throughout the BiOI-0.25-60 NSs, which is consistent with the XPS results.

Figure 4a,b and the inset in Figure 4b display the TEM, HRTEM, and SAED images of BiOI-0.25-60 NSs. Figure 4a clearly reveals the layered structure of the BiOI-0.25-60 NSs. HRTEM analysis shows a crystalline spacing of 0.304 nm, corresponding to the (102) lattice plane of tetragonal phase BiOI [69]. The inset electron diffraction pattern indicates that BiOI-0.25-60 is a single crystal structure.

### 3.2. Optical and Electrical Properties

The optical absorption properties of BiOI-0.25-60 NSs and BiOI-0-70 NSs were determined by DRS. As illustrated in Figure 5a, the absorption edges of the BiOI-0.25-60 NSs and BiOI-0-70 NSs are approximately 510 nm and 550 nm, respectively, indicating good visible light absorption for both samples. The generation, separation, and migration capabilities of photogenerated charges in the synthesized BiOI-0.25-60 NSs and BiOI-0-70 NSs were examined using PL, i-t, and EIS measurements, as shown in Figure 5b–d. According to Figure 5b, the PL intensity of the BiOI-0.25-60 NS sample is weaker and red-shifted compared to BiOI-0-70 NSs, indicating that the BiOI samples prepared with the addition of mannitol were able to reduce the recombination of photogenerated carriers [70]. Figure 5c shows that the photocurrent intensity of the BiOI-0.25-60 NSs is significantly higher than that of BiOI-0-70 NSs and remains consistently stable, suggesting more effective separation of electron-hole pairs in BiOI-0.25-60 NSs. As shown in Figure 5d, the EIS data for BiOI-0.25-60 NSs and BiOI-0-70 NSs indicate that BiOI-0.25-60 NSs exhibit the smallest semicircular arc radius in the Nyquist plot, signifying reduced charge transfer resistance and thus superior conductivity. These results indicate that the prepared BiOI-0.25-60 NSs have excellent photoelectrochemical properties and charge separation efficiency.

### 3.3. Photocatalytic Degradation Properties

Figure 6a displays the photocatalytic degradation curves of the BiOI NSs with different amounts of mannitol for the degradation of 10 mg/L MB solution under simulated sunlight. After 100 min, the auto-degradation efficiency of MB was only 1.7%, indicating a negligible effect on the photocatalytic activity of the photocatalysts. As the mannitol dosage increased from 0 g (BiOI-0.1-70) to 0.25 g (BiOI-0.25-70), the photocatalytic efficiency of the BiOI photocatalyst for MB degradation improved significantly, rising from 46.2% to 84.8%. However, further increases in mannitol dosage resulted in a decline in photocatalytic efficiency, which dropped to 77.1% at a dosage of 0.35 g (BiOI-0.35-70). Together with the BET, SEM, and photoelectrochemical performance results, these findings suggest that an optimal amount of mannitol effectively modulates the size, specific surface area, charge carrier separation, and transport properties of BiOI, thereby enhancing its photocatalytic performance. The Langmuir–Hinshelwood model was used to translate the experimental data, which can be described by the equation *ln(C/C*_0_*) = −kt* [71], where the rate constant (k) serves as an efficiency indicator. Figure 6b displays the first-order kinetic characteristics of BiOI-0-70, BiOI-0.1-70, BiOI-0.15-70, BiOI-0.2-70, BiOI-0.25-70, BiOI-0.3-70, and BiOI-0.35-70 NSs for RhB degradation, with respective rate constants of 0.004 min^−1^, 0.007 min^−1^, 0.008 min^−1^, 0.008 min^−1^, 0.013 min^−1^, 0.011 min^−1^, and 0.009 min^−1^. BiOI-0.25-70 NSs showed the quickest degradation rate, approximately 3.25 times that of BiOI-0-70 NSs.

Figure 6c,d present the photocatalytic degradation curves and first-order kinetic curves for MB degradation using BiOI-0.25-40, BiOI-0.25-50, BiOI-0.25-60, BiOI-0.25-70, and BiOI-0.25-80 NSs under varied temperature conditions. As shown in Figure 6c, after 100 min of simulated solar irradiation, the degradation efficiencies were 91.7%, 89.4%, 96.5%, 84.8%, and 73.8%, respectively. Among these, BiOI-0.25-60 NSs exhibited the highest photocatalytic performance, with a degradation efficiency 1.14 times greater than that of BiOI-0.25-70 NSs. Figure 6d highlights the corresponding first-order rate constants for MB degradation, which were determined to be 0.017 min^−1^, 0.015 min^−1^, 0.024 min^−1^, 0.013 min^−1^, and 0.008 min^−1^ for BiOI-0.25-40, BiOI-0.25-50, BiOI-0.25-60, BiOI-0.25-70, and BiOI-0.25-80 NSs, respectively. Notably, the BiOI-0.25-60 NSs demonstrated the fastest deterioration rate, 1.84 times that of BiOI-0-70 NSs. The findings confirm that BiOI-0.25-60 NSs exhibit outstanding photocatalytic degradation capabilities, outperforming the other samples under the investigated conditions.

To evaluate the effect of aqueous matrix on the photocatalytic degradation of MB by BiOI-0.25-60 NSs, tap water and mineral water were utilized to prepare MB solutions. As shown in Figure 6e, the degradation efficiencies of MB in tap water and mineral water reached 99.8% and 98.5%, respectively. These values are comparable to the efficiency observed in deionized water (96.5%), suggesting that the presence of cations and anions in tap and mineral water may have a slight promoting effect on the degradation process [72]. Figure 6f further demonstrates the sensitization effect of BiOI-0.25-60 NSs, as evidenced by their ability to degrade various pollutants. The degradation efficiencies achieved for TC, 2,4-D, and Rh B were 89.6%, 27.8%, and 84.8%, respectively.

To explore the influence of key parameters, including photocatalyst loading, pollutant concentration, and pH value, on catalytic performance, a series of experiments were conducted using the BiOI-0.25-60 NSs, which demonstrated optimal photocatalytic activity. Figure 6g shows that the degradation efficiency increased significantly as the photocatalyst loading was raised from 100 mg/L to 300 mg/L at an MB concentration of 10 mg/L. This trend indicates a positive correlation between photocatalyst mass and degradation efficiency. Figure 6h illustrates that the degradation efficiency gradually decreased as the MB concentration increased from 5 mg/L to 15 mg/L. This suggests that higher pollutant concentrations impose greater demands on the catalyst, thereby limiting its ability to efficiently degrade the pollutant due to saturation of active sites or light attenuation effects. To study the effect of pH, HCl or NaOH were used to adjust the acidity or alkalinity of the MB solution, respectively. Figure 6i depicts the degradation curves across a range of pH values, revealing the significant role of pH in modulating photocatalytic performance. The results showed that the degradation efficiency was slightly enhanced under acidic (pH = 4) and basic (pH = 8) conditions. At lower pH levels, the acidic environment caused the BiOI surface to become positively charged, while MB molecules carried a negative charge. This charge difference resulted in stronger electrostatic attraction between the photocatalytic surface and MB molecules, promoting their adsorption and contact with active sites, thereby accelerating the photocatalytic degradation process. At higher pH levels, the surface charge properties of BiOI changed, potentially leading to a weak negative charge on the catalyst surface. In addition, under alkaline conditions, partial dissociation of MB molecules occurred, further increasing the reactivity between the photocatalytic and MB. Moreover, the higher OH^−^ concentration in an alkaline environment likely generated more hydroxyl radicals (·OH), which are highly oxidative species that play a crucial role in the photocatalytic degradation process, thereby enhancing the degradation efficiency of MB [73]. Figure 6j displays the cycling performance of BiOI-0.25-60 NSs. After three consecutive cycles, the photocatalytic degradation efficiency of BiOI-0.25-60 was still above 95.3%, showing only a minor decline of 1.2% compared to the initial performance. This result confirms the excellent stability and recyclability of BiOI-0.25-60 NSs during the photocatalytic degradation of MB.

### 3.4. Photocatalytic Degradation Mechanism

Figure 7a depicts the Tauc plots of BiOI-0.25-60 NSs and BiOI-0-70 NSs, allowing the calculation of their bandgap energy (*E_g_*) based on the following equation:*(αhv)^2^ = K(hv − E_g_)*(3)
where *K* is a material-dependent parameter; *h* represents Planck’s constant; *α* is the absorption coefficient; and *v* is the incident photon frequency. As shown in Figure 7a, the calculated *E_g_* of BiOI-0.25-60 NSs and BiOI-0-70 NSs are 2.28 eV and 2.12 eV, respectively.

The flat band (Fermi energy levels, *E_F_*) of the BiOI-0.25-60 NSs and BiOI-0-70 NSs were determined using M-S curves. As demonstrated in Figure 7b,c, the *E_F_* potentials of BiOI-0.25-60 NSs and BiOI-0-70 NSs are 0.91 eV and −0.13 eV vs. Ag/AgCl, respectively. These values can be converted to the standard hydrogen electrode (NHE) potentials utilizing the following equation:*E_NHE_ = E_Ag__/AgCl_ + 0.198 V*(4)

Accordingly, the *E_F_* potentials of BiOI-0.25-60 NSs and BiOI-0-70 NSs were determined to be 1.108 and 0.068 V vs. NHE, respectively. Moreover, a negative slope of the M-S curve for BiOI-0.25-60 NSs confirms its p-type semiconductor characteristics, while the positive slope observed for BiOI-0-70 NSs indicates its n-type semiconductor nature [74]. This suggests that the addition of the mannitol active agent induces a transition in BiOI from an n-type to a p-type semiconductor. As shown in Figure 7d, the relative potentials of valence band (VB) vs. *E_F_* for BiOI-0.25-60 NSs and BiOI-0-70 NSs were derived from VB-XPS spectral analysis, yielding values of 0.35 eV and 1.40 eV, respectively. Consequently, the valence band position (*E_VB_*) of BiOI-0.25-60 NSs and BiOI-0-70 NSs were calculated as 1.458 eV and 1.468 eV, respectively. And the conduction band position (*E_CB_*) of them were calculated using Equation (5), as follows:*E_vb_ = E_cb_ + E_g_*(5)

Based on this relationship, the *E_CB_* values for BiOI-0.25-60 NSs and BiOI-0-70 NSs were calculated to be −0.822 V and −0.652 V vs. NHE, respectively. From the above calculations, it can be inferred that VB of BiOI-0.25-60 NSs is closer to its *E_F_*, confirming its p-type semiconductor characteristics [75]. Conversely, the CB of BiOI-0-70 NSs is located near its *E_F_*, indicating n-type semiconductor behavior [76]. The transition of BiOI from n-type to p-type semiconductor behavior is induced by the introduction of mannitol during the synthesis process, which has significant implications for photocatalytic performance. Mannitol, as an additive, may selectively generate oxygen vacancies at specific sites in BiOI-0.25-60. The p-type semiconductor characteristics of BiOI are significantly influenced by the synthesis environment, particularly intrinsic defects such as oxygen vacancies and iodine defects. This defect engineering introduces a new energy level between the bandgap and the valence band [77,78], facilitating the transition from n-type to p-type semiconductor properties. This transition increases the concentration of holes (h^+^) as the primary charge carriers, significantly enhancing photocatalytic activity and charge transfer efficiency.

Based on the results of the analysis presented above, the energy band positions of BiOI-0.25-60 NSs and BiOI-0-70 NSs and their photocatalytic degradation mechanism under simulated sunlight are illustrated in Figure 8a. When exposed to the simulated sunlight, the BiOI-0.25-60 NSs and BiOI-0-70 NSs absorb the photons, causing electrons (e^−^) to transition from the VB to the CB, leaving corresponding h^+^ on the VB. The h^+^ remaining in the VB exhibits strong oxidative properties and can directly decompose pollutants into H_2_O and CO_2_. Additionally, the CB positions of BiOI-0.25-60 NSs and BiOI-0-70 NSs are −0.822 eV and −0.625 eV vs. NHE, respectively, both more negative than the energy level of O_2_/superoxide anion radicals (∙O^2−^) (−0.33 eV vs. NHE). This allows e^−^ to migrate to the surface of the samples and react with O_2_, generating ∙O^2−^. These reactive species further degrade pollutants, ultimately converting them into harmless H_2_O and CO_2_. Notably, the CB position of BiOI-0.25-60 NSs is significantly more negative than that of BiOI-0-70 NSs, endowing the electrons in the CB of BiOI-0.25-60 NSs with a stronger reduction potential. This enhanced reduction ability facilitates the formation of a greater quantity of reactive ∙O^2−^ species. However, as the VB positions of both BiOI-0.25-60 NSs and BiOI-0-70 NSs are lower than the oxidation potential of ∙OH/H_2_O (2.34 V vs. NHE), h^+^ cannot directly convert H_2_O into hydroxyl radicals (∙OH). Nevertheless, the presence of ∙O^2−^ enables further reactions with H_2_O, generating small amounts of ∙OH, which also contribute to pollutant degradation. In summary, BiOI-0.25-60 NSs demonstrate superior photocatalytic activity compared to BiOI-0-70 NSs, attributed to its smaller nanoscale size, larger specific surface area, enhanced optoelectronic properties, more negative CB position, and optimal band gap width.

To provide direct evidence of the active species involved in the photocatalytic mechanism described above, the degradation efficiency of MB degradation by BiOI-0.25-60 NSs was investigated using various scavengers. As displayed in Figure 8b, the addition of EDTA-2Na and TEMPO, which serve as trapping agents for h^+^ and ·O^2−^, reduced the degradation rate of MB from 96.5% (without scavengers) to 67.3% and 24.0%, respectively. This result demonstrated that h^+^ and ·O^2−^ are the primary active species, with ·O^2−^ playing a particularly significant role. In contrast, the use of 1 mM of IPA as a ·OH trapping agent resulted in a minimal decrease in MB degradation efficiency, from 96.5% to 94.1%, suggesting that ·OH has a negligible impact on MB degradation, which is consistent with photocatalytic mechanism proposed in this study.

In recent years, researchers have conducted various photocatalytic studies based on BiOI. As shown in Table 1, although all of these studies focus on the photocatalytic degradation of pollutants by BiOI and its different configurations, the degradation efficiencies vary due to differences in experimental conditions. These factors include the type and dosage of photocatalyst, the type and concentration of pollutants, the intensity and type of light source used, exposure time, and the distance between the sample and the light source.

## 4. Conclusions

In this study, a series of BiOI NSs were successfully synthesized with tunable structural and electronic properties by varying the mannitol concentration and reaction temperatures. The introduction of mannitol was found to induce a transformation in BiOI NSs from an n-type to a p-type semiconductor, as confirmed by M-S curves and band structure analyses. This transition increased the availability of ·O^2−^ as primary charge carriers, significantly enhancing charge transfer efficiency and photocatalytic activity. Among the synthesized samples, BiOI-0.25-60 NSs demonstrated superior performance, achieving 96.5% MB degradation efficiency under simulated sunlight. Notably, the photocatalytic efficiency remained high across the different aqueous matrices of 99.8% in tap water and 98.5% in mineral water, suggesting a slight promoting effect of cations and anions. The BiOI-0.25-60 NSs also effectively degraded multiple pollutants, with efficiencies of 89.6% for TC, 27.8% for 2,4-D, and 84.8% for Rh B. Key experimental parameters influenced photocatalytic performance, including catalyst dosage, pollutant concentration and pH. Higher photocatalyst dosages enhanced efficiency, while increasing MB concentrations reduced it due to active site saturation and light attenuation. pH played a critical role, with acidic (pH = 4) and basic (pH = 8) conditions enhancing efficiency via electrostatic interactions and increased ·OH generation. Scavenger experiments further validated ·O^2−^ and h^+^ as the primary reactive species, with ·OH playing a negligible role. These findings emphasize the importance of mannitol in modulating the semiconductor properties of BiOI nanomaterials, providing valuable insights for developing advanced p-type photocatalysts for environmental remediation.

## Figures and Tables

**Figure 1 nanomaterials-14-02048-f001:**
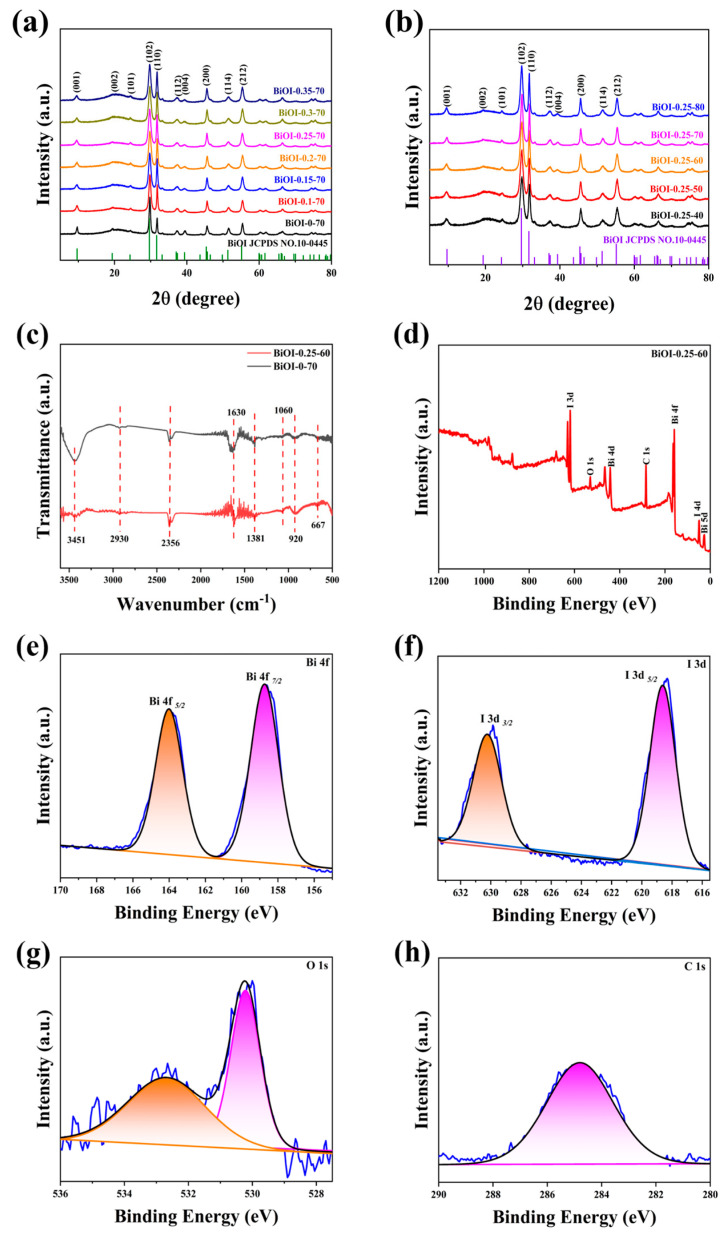
(**a**) XRD patterns of BiOI synthesized with different amounts of mannitol; (**b**) BiOI synthesized with 0.25 g of mannitol at different temperatures; (**c**) FT-IR spectra of BiOI-0.25-60 NSs and BiOI-0-70 NSs; (**d**) XPS survey spectrum of BiOI-0.25-60 NSs; (**e**–**h**) high-resolution XPS spectra for Bi, I, O, and C elements.

**Figure 2 nanomaterials-14-02048-f002:**
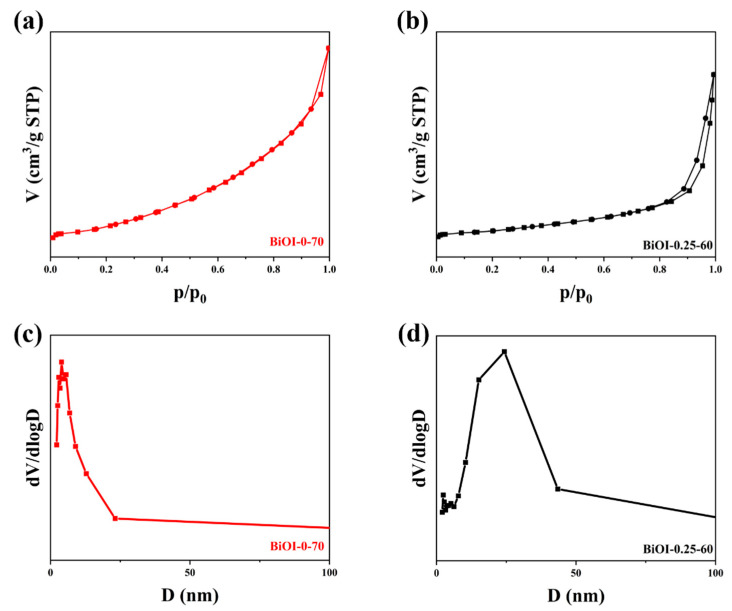
N_2_ adsorption−desorption isotherms (**a**,**b**) and pore diameter distributions (**c**,**d**) of BiOI-0-70 NSs and BiOI-0.25-60 NSs.

**Figure 3 nanomaterials-14-02048-f003:**
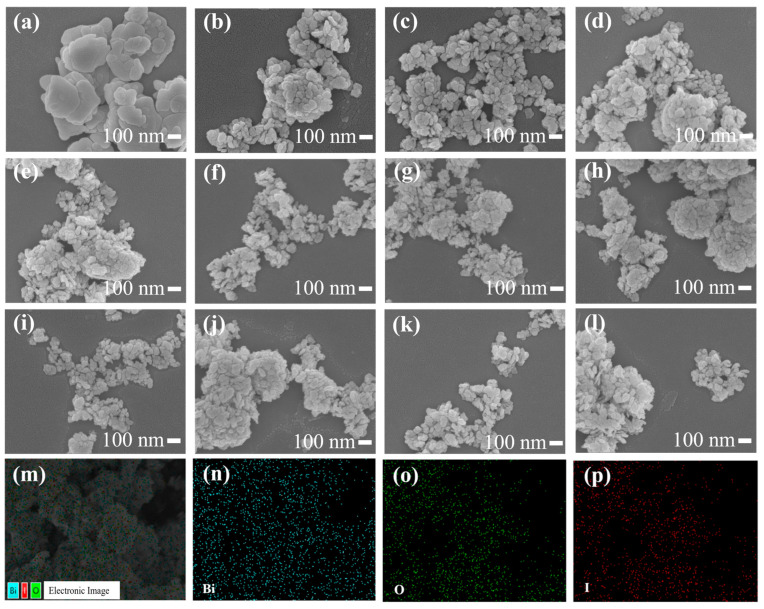
SEM images of (**a**) BiOI-0-70; (**b**) BiOI-0.1-70; (**c**) BiOI-0.15-70; (**d**) BiOI-0.2-70; (**e**) BiOI-0.25-70; (**f**) BiOI-0.3-70; (**g**) BiOI-0.35-70; (**h**) BiOI-0.25-40; (**i**) BiOI-0.25-50; (**j**) BiOI-0.25-60; (**k**) BiOI-0.25-70; (**l**) BiOI-0.25-80; and (**m**–**p**) EDS maps of BiOI-0.25-60.

**Figure 4 nanomaterials-14-02048-f004:**
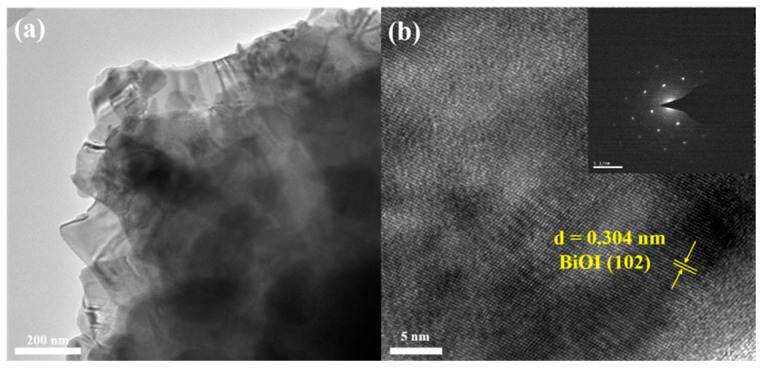
(**a**,**b**) TEM and HRTEM images of BiOI-0.25-60 NSs. The inset in (**b**) displays the SAED pattern.

**Figure 5 nanomaterials-14-02048-f005:**
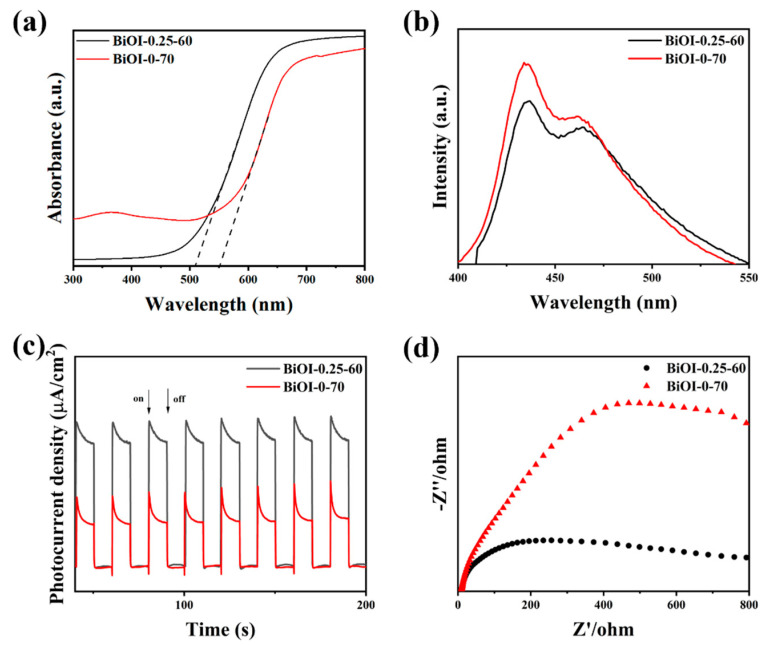
(**a**) UV-Vis diffuse reflectance spectra; (**b**) PL spectra; (**c**) i-t curves; (**d**) EIS spectra of BiOI-0.25-60 NSs and BiOI-0-70 NSs.

**Figure 6 nanomaterials-14-02048-f006:**
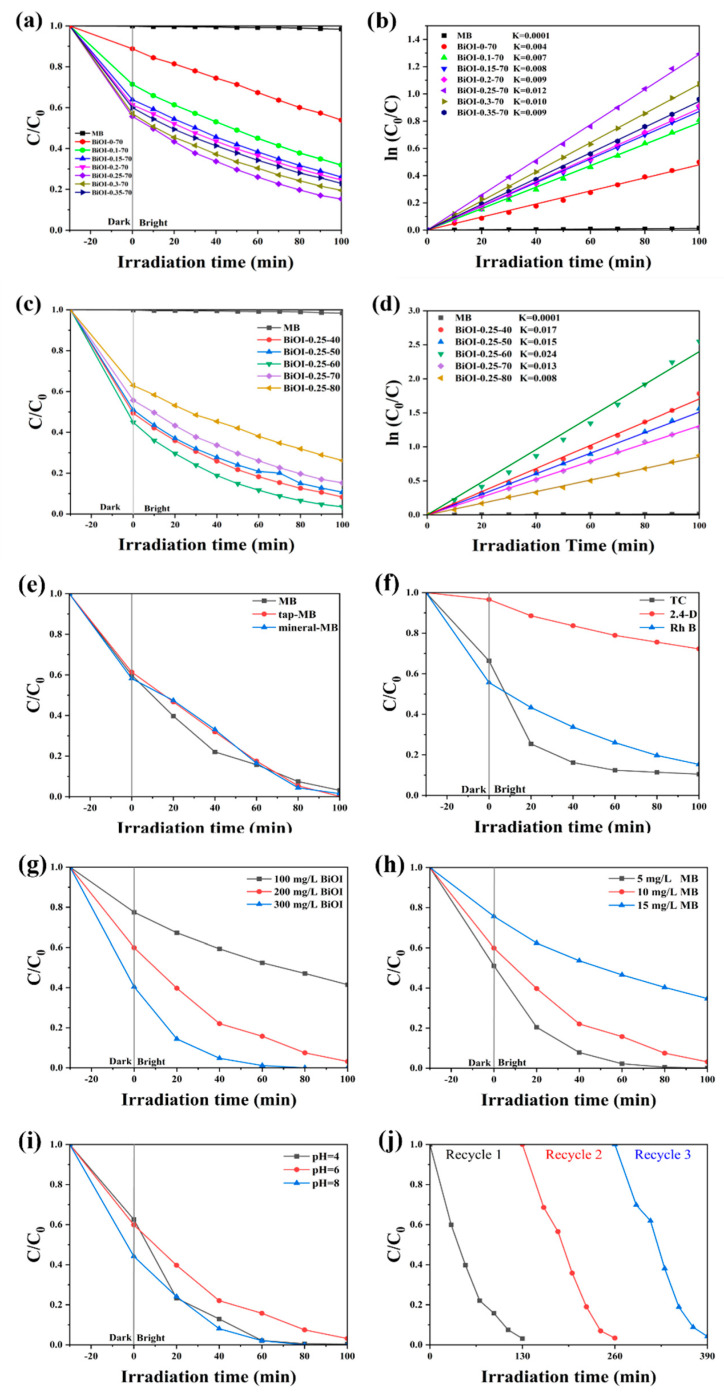
(**a**,**c**) Photocatalytic degradation efficiency and (**b**,**d**) first-order kinetics of each sample for the degradation of MB under simulated sunlight as a function of time; for BiOI-0.25-60 NSs, photocatalytic degradation efficiency is shown under various conditions: (**e**) degradation of MB in deionized water, tap water, and mineral water; (**f**) degradation of TC, 2,4-D, and RhB; (**g**) different dosages of BiOI-0.25-60 NSs; (**h**) different MB concentrations; (**i**) different pH values; (**j**) cycling runs of BiOI-0.25-60 for MB degradation.

**Figure 7 nanomaterials-14-02048-f007:**
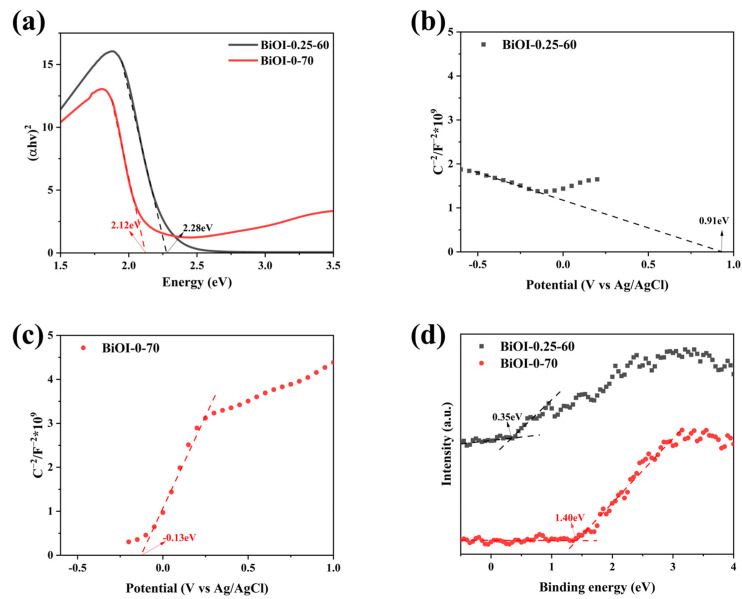
(**a**) Tauc plots; (**b**,**c**) M-S curves; (**d**) VB-XPS spectra of BiOI-0.25-60 NSs and BiOI-0-70 NSs.

**Figure 8 nanomaterials-14-02048-f008:**
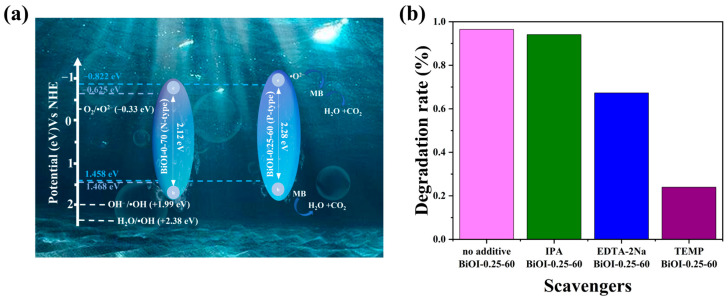
(**a**) Photocatalytic mechanism for the degradation of MB by BiOI-0.25-60 NSs and BiOI-0-70 NSs under simulated sunlight; (**b**) effect of different scavengers on the photocatalytic degradation of MB by BiOI-0.25-60 NSs.

**Table 1 nanomaterials-14-02048-t001:** List of BiOI configurations photocatalysts studied for degrading various pollutants.

Sample	Amount	Application	Concentration and Usage	Power Source	Time	Efficiency	Ref.
BiOI	25 mg	gallic acid degradation	20 ppm 250 mL	Xenion lamp 12 W	60 min	60%	[46]
BiOI	50 mg	NOR degradation	10 mg/L 200 mL	Panasonic lamp 15W	240 min	80%	[79]
BiOI heterojunction	10 mg	TC degradation	20 mg/L 10 mL	200 W	120 min	69.43%	[80]
N doped BiOI	50 mg	TC degradation	20 mg/L 60 mL	500 W	120 min	70%	[81]
Ag/BiOI	50 mg	MO degradation	10 mg/L 50 mL	Xe lamp 500 W	240 min	80%	[82]
g-C_3_N_4_/BiOI	20 mg	TC degradation	10 mg/L 50 mL	Xe lamp 500 W	120 min	77%	[83]
PANI/BiOI	20 mg	RhB degradation	20 mg/L 50 mL	Xe lamp 300 W	120 min	91%	[45]
rGO/BiOI	80 mg	MO degradation	10 mg/L 80 mL	Xe lamp 250 W	240 min	85%	[84]
Pt/BiOI	50 mg	AO II	20 mg/L 80 mL	Tungsten halogen lamp 300 W	60 min	90%	[85]
Bi/BiOI	5 mg	BPA	20 mg/L 10 mL	Xe lamp 350 W	60 min	90%	[86]
BiOI/BiOBr/Bi	20 mg	TC	10 mg/L 50 mL	Xe lamp 300 W	140 min	98.4%	[87]

## Data Availability

Data are contained within the article.

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
