# Peer review of "Engineering n-Type and p-Type BiOI Nanosheets: Influence of Mannitol on Semiconductor Behavior and Photocatalytic Activity"

_nanomaterials, 2024, doi:10.3390/nano14242048_

Round 1
Reviewer 1 Report
Comments and Suggestions for Authors
This manuscript proposes the use of a Bi-based nanostructured which is "doped" with mannitol, in order to improve the photocatalytic performance towards MB degradation. Though the use of mannitol has been extensively reported, the authors are focusing on a specific application, trying to describe the changes at the electronic structure level. I found the manuscript well written, and I suggest the authors to address the following points:
1. Please check the grammar and overall text for typos, since I was able to find a couple of errors.
2. Please correct the references where the Chinese characters appear.
3. I suggest the authors to elaborate more on the use of mannitol as a "doping" molecule. How other similar molecules (such as sorbitol) could impact in the photocatalytic activity?
Author Response
Comments 1: Please check the grammar and overall text for typos, since I was able to find a couple of errors. |
Response 1: Thank you for your valuable feedback and patience in pointing out the errors. I have carefully reviewed the text and corrected the identified issues. Below is the revised version, which I believe is now more accurate. If there are any additional concerns, please do not hesitate to let me know. In line 191, "are" has been corrected to "is." In line 215, "classifying" has been changed to "indicating." In line 246, "edging" has been revised to "edges." In line 252, "complexation" has been corrected to "recombination." In line 372, " On the basis of " has been updated to "Base on." In line 388, "demonstrates" has been updated to "demonstrate."
|
Comments 2: Please correct the references where the Chinese characters appear. |
Response 2: Thank you very much for your valuable feedback on the manuscript. I would like to apologize for the inconvenience caused by the appearance of Chinese characters in the references section, which resulted from encoding errors within our submission system. Please rest assured that I have thoroughly reviewed and corrected these issues. I will take extra precautions to ensure this does not recur in the revised submission. Thank you for your understanding and your continued guidance. Incorrectly formatted references have been re-cited and reorganized, especially references 57-66 on lines 185-203.
Comments 3: I suggest the authors to elaborate more on the use of mannitol as a "doping" molecule. How other similar molecules (such as sorbitol) could impact in the photocatalytic activity? Response 3: Thank you very much for your suggestion. We fully acknowledge the importance of a more in-depth discussion on the use of mannitol and other similar molecules in enhancing our understanding of photocatalytic activity. In response to your feedback, we have significantly expanded this discussion in the introduction of the revised manuscript. A comprehensive analysis detailing how mannitol and other similar molecules impact photocatalytic performance is now thoroughly incorporated, with specific explanations found from lines 83 to 98 and 361 to 367.
|
Reviewer 2 Report
Comments and Suggestions for Authors
Section 3.1, check the XRD, FTIR & XPS discussion parts for the proper citations.
Figure 3 (a-j), the scale bars of the SEM images are not visible enough.
Photodegradation of MB needs to be evaluated under different parameters such as (i) different catalyst loadings (ii) different pH (iii) different MB concentrations
The photocatalytic performance of the synthesized catalysts further needs to be confirmed with colourless pollutant such as phenol.
Abstract and conclusions needs to be improved by stating important findings/results of this study.
Author Response
Comments 1: Section 3.1, check the XRD, FTIR & XPS discussion parts for the proper citations.
Response 1: Thank you for your attention to detail and for pointing out the need for proper citations in the discussion sections of XRD, FTIR, and XPS in Section 3.1 of our manuscript. We have carefully reviewed these sections and updated them with the appropriate citations to ensure that all data and statements are accurately referenced. We appreciate your guidance in helping us improve the quality and accuracy of our manuscript. Incorrectly formatted references have been re-cited and reorganized, especially references 57-66 on lines 185-203.
Comments 2: Figure 3 (a-j), the scale bars of the SEM images are not visible enough.
Response 2: Thank you for pointing out the visibility issue with the scale bars in Figure 3 (a-j) of our manuscript. We acknowledge the importance of clearly visible scale bars for the accurate interpretation of SEM images. We have revised the images to enhance the visibility of the scale bars, ensuring that they are now clearly distinguishable and appropriately scaled for each image. This revision will facilitate a better understanding of the image details and improve the overall presentation of the data. Thank you again for your constructive feedback.
Comments 3: Photodegradation of MB needs to be evaluated under different parameters such as (i) different catalyst loadings (ii) different pH (iii) different MB concentrations.
Response 3: Thank you for your suggestions regarding the evaluation of the photodegradation of methylene blue (MB) under various parameters. We appreciate your input and recognize the importance of these factors in assessing the efficiency of our photocatalysts. We are pleased to inform you that we have already conducted experiments to assess the effects of different catalyst loadings, pH levels, and MB concentrations on the photodegradation process. The results of these experiments are presented in Figure 6 (g-i) of the revised manuscript. For a detailed explanation and discussion of these findings, please refer to lines 301-323 in the revised manuscript. These additional experiments have provided valuable insights into the optimal conditions for the photodegradation of MB, enhancing our understanding of the catalyst's performance under varied environmental conditions. Thank you once again for your constructive feedback which has helped to strengthen our study
Comments 4: The photocatalytic performance of the synthesized catalysts further needs to be confirmed with colourless pollutant such as phenol.
Response 4: Thank you for your suggestion to evaluate the photocatalytic performance of our synthesized catalysts using colorless pollutants. We understand the importance of testing a broad spectrum of contaminants to comprehensively assess the capabilities of our photocatalysts. We are pleased to inform you that we have already conducted photocatalytic degradation experiments using colorless pollutants, specifically tetracycline (TC) and 2,4-dichlorophenol (2,4-D). The results of these experiments are detailed in Figures 6(f) of the revised manuscript. For a thorough explanation and discussion of these findings, please refer to lines 298-300 in the revised manuscript. We hope this additional data will further validate the effectiveness of our catalysts under varied conditions and with different types of pollutants. Thank you again for your valuable feedback, which has helped enhance the rigor of our study.
Comments 5: Abstract and conclusions needs to be improved by stating important findings/results of this study.
Response 5: Thank you for your valuable feedback on the abstract and conclusions of our manuscript. We appreciate your guidance in emphasizing the importance of highlighting key findings and results in these sections. In response to your suggestion, we have revised both the abstract and the conclusions to more clearly articulate the significant findings and implications of our study. The updated abstract now succinctly summarizes the major outcomes, while the revised conclusions provide a more comprehensive reflection on the implications of these results and their relevance to the field. We hope these revisions will improve the readability and impact of our manuscript, making the principal contributions of our research more accessible to readers. Thank you once again for your constructive comments that have helped enhance our manuscript's clarity and focus.
Reviewer 3 Report
Comments and Suggestions for Authors
MS No: |
nanomaterials-3354174 |
Title:
|
Engineering n-Type and p-Type BiOI Nanosheets: Influence of mannitol on Semiconductor Behavior and Photocatalytic Activity |
Authors: |
Shuo Yang, Wenhui Li, Kaiyue Li, Ping Huang, Yuquan Zhuo, Keyan Liu, Ziwen Yang, Donglai Han |
The present manuscript deals with the synthesis of BiOI nanosheets (NSs) using a precipitation technique with varying amounts of mannitol. The photocatalytic materials were characterized in detail considering their physicochemical properties and tested for methylene blue degradation in ultrapure water. In general, while the preparation method is interesting, the use of methylene blue as the target pollutant lowers its scientific value as due to photosensitization phenomena, false conclusions may occur. In my opinion, it can be accepted for publication in Nanomaterials after major revision.
Below are some specific comments.
· Avoid using three decimal numbers in the surface area BET measurements. Two are ok.
· Correct the references format in the text.
· Additional data considering the degradation of emerging pollutants (e.g pharmaceuticals, parabens, bisphenol etc) should be added to the revised manuscript.
· The stability of the as prepared materials should be shown in the revised manuscript through consecutive experimental runs.
· The effect of water matrix (both in synthetic and real matrices) should be added to the revised manuscript.
· The Authors state that they used a 500W xenon lamp for the photocatalytic tests. I would advise them to perform a chemical actinometry in order to find out the light intensity entering their photoreactor.
· A Table comparing the efficiency of the prepared materials with other BiOI configurations should be added to the revised manuscript.
Author Response
Comments 1: Avoid using three decimal numbers in the surface area BET measurements. Two are ok. |
||||||||||||||||||||||||||||||||||||||||||||||||||||||||||||||||||||||||||||||||||||||||||||||||
Response 1: Thank you for your valuable suggestions and attention to the details of the manuscript. We fully understand your guidance on the presentation of data from specific surface area BET measurements. We have made sure to round the surface area BET measurements to two decimal places in the revised manuscript, as recommended. The specific modifications can be found on line 211-215 of the revised manuscript. And in line 27, "24.462" has been corrected to "24.46."
|
||||||||||||||||||||||||||||||||||||||||||||||||||||||||||||||||||||||||||||||||||||||||||||||||
Comments 2: Correct the references format in the text. |
||||||||||||||||||||||||||||||||||||||||||||||||||||||||||||||||||||||||||||||||||||||||||||||||
Response 2: I would like to apologize for the inconvenience caused by the appearance of Chinese characters in the references section, which resulted from encoding errors within our submission system. Please rest assured that I have thoroughly reviewed and corrected these issues. I will take extra precautions to ensure this does not recur in the revised submission. Thank you for your understanding and your continued guidance. Incorrectly formatted references have been re-cited and reorganized, especially references 57-66 on lines 185-203.
Comments 3: Additional data considering the degradation of emerging pollutants (e.g pharmaceuticals, parabens, bisphenol etc) should be added to the revised manuscript. Response 3: Thank you for your insightful suggestion to include additional data on the degradation of emerging pollutants. We have already conducted photocatalytic degradation experiments on tetracycline (TC), 2,4-dichlorophenol (2,4-D) and Rhodamine B (Rh B). The specific data from these experiments are presented in Figure 6 (f) of the revised manuscript. For further details and explanations, please refer to lines 298-300 of the revised manuscript. We hope these additions address your concerns adequately.
Comments 4: The stability of the as prepared materials should be shown in the revised manuscript through consecutive experimental runs. Response 4: Thank you for your constructive suggestion regarding the stability demonstration of the materials prepared. We have addressed this comment by performing cyclic experiments on methylene blue (MB). After three consecutive cycles, the photocatalytic degradation efficiency of BiOI-0.25-60 was still 95.3%, only showing a decrease of 1.2% from the initial performance, which confirms the good cyclic stability of our material. Detailed explanations and results of these experiments can be found on lines 323-327 of the revised manuscript.
Comments 5: The effect of water matrix (both in synthetic and real matrices) should be added to the revised manuscript. Response 5: Thank you for your insightful suggestion. We fully agree on the importance of including data on the effects of different aqueous matrices on the material properties in our revised manuscript. To address this, we conducted additional experiments using tap water and mineral water as solvents for a 10 mg/L methylene blue (MB) solution, to evaluate the degradation efficiency of BiOI nanosheets in these matrices. The experiments showed that the degradation efficiency in tap water (Tap-MB) could reach 99.8%, and in mineral water (Mineral-MB), it could reach 98.5%. These results indicate that the photocatalytic degradation efficiency of the BiOI nanosheets did not decrease but rather showed a slight improvement, possibly due to a minor influence of the ions in the water on the degradation process [1]. These experiments underscore the strong practical application potential and effectiveness of our prepared materials under different aquatic environmental conditions. The specific data from these experiments are presented in Figure 6(e) of the revised manuscript, and detailed explanations can be found on lines 293-298. Ref: [1] Tang, W.; Chen, J.; Yin, Z.; Sheng, W.; Lin, F.; Xu, H.; Cao, S., Complete removal of phenolic contaminants from bismuth-modified TiO2 single-crystal photocatalysts. Chinese Journal of Catalysis 2021, 42 (2), 347-355.
Comments 6: The Authors state that they used a 500W xenon lamp for the photocatalytic tests. I would advise them to perform a chemical actinometry in order to find out the light intensity entering their photoreactor. Response 6: We sincerely appreciate your valuable suggestions. Following your advice, we measured and calculated the luminous flux of the photoreactor. Due to the limitations of laboratory conditions, we used an illuminance meter for the measurements and, through indirect calculations, determined that the luminous flux at the liquid surface inside the photoreactor is approximately 630 lumens. Once again, we thank the reviewer for the thorough review of our work.
Comments 7: A Table comparing the efficiency of the prepared materials with other BiOI configurations should be added to the revised manuscript. Response 7: Thank you for your suggestion. We have taken your advice and added a table comparing the efficiency of our prepared materials with other BiOI configurations to the revised manuscript. This table is included as Table 1. For further details and explanations regarding these comparisons, please refer to lines 410 of the revised manuscript.
Table 1 List of BiOI configurations photocatalysts studied for degrading various pollutants.
|
||||||||||||||||||||||||||||||||||||||||||||||||||||||||||||||||||||||||||||||||||||||||||||||||
Ref: [2] Mera, A. C.; Moreno, Y.; Contreras, D.; Escalona, N.; Meléndrez, M. F.; Mangalaraja, R. V.; Mansilla, H. D., Improvement of the BiOI photocatalytic activity optimizing the solvothermal synthesis. Solid State Sciences 2017, 63, 84-92. [3] Narenuch, T.; Senasu, T.; Chankhanittha, T.; Nanan, S., Sunlight-active BiOI photocatalyst as an efficient adsorbent for the removal of organic dyes and antibiotics from aqueous solutions. Molecules 2021, 26 (18), 5624. [4] Sun, Y.; Ahmadi, Y.; Younis, S. A.; Kim, K.-H., Modification strategies of BiOI-based visible-light photocatalysts and their efficacy on decomposition of tetracycline antibiotics in water. Critical Reviews in Environmental Science and Technology 2024, 1-30. [5] Ma, F.-Q.; Yao, J.-W.; Zhang, Y.-F.; Wei, Y., Unique band structure enhanced visible light photocatalytic activity of phosphorus-doped BiOI hierarchical microspheres. RSC advances 2017, 7 (58), 36288-36296. [6] Liu, H.; Cao, W.; Su, Y.; Wang, Y.; Wang, X., Synthesis, characterization and photocatalytic performance of novel visible-light-induced Ag/BiOI. Applied Catalysis B: Environmental 2012, 111, 271-279. [7] Di, J.; Xia, J.; Yin, S.; Xu, H.; Xu, L.; Xu, Y.; He, M.; Li, H., Preparation of sphere-like gC3N4/BiOI photocatalysts via a reactable ionic liquid for visible-light-driven photocatalytic degradation of pollutants. Journal of Materials Chemistry A 2014, 2 (15), 5340-5351. [8] Arumugam, M.; Choi, M. Y., Recent progress on bismuth oxyiodide (BiOI) photocatalyst for environmental remediation. Journal of Industrial and Engineering Chemistry 2020, 81, 237-268. [9] Vinoth, R.; Babu, S. G.; Ramachandran, R.; Neppolian, B., Bismuth oxyiodide incorporated reduced graphene oxide nanocomposite material as an efficient photocatalyst for visible light assisted degradation of organic pollutants. Applied Surface Science 2017, 418, 163-170. [10] Yu, C.; Jimmy, C. Y.; Fan, C.; Wen, H.; Hu, S., Synthesis and characterization of Pt/BiOI nanoplate catalyst with enhanced activity under visible light irradiation. Materials Science and Engineering: B 2010, 166 (3), 213-219. [11] Chang, C.; Zhu, L.; Fu, Y.; Chu, X., Highly active Bi/BiOI composite synthesized by one-step reaction and its capacity to degrade bisphenol A under simulated solar light irradiation. Chemical engineering journal 2013, 233, 305-314. [12] Wang, X.; Liang, H.; Zhao, X.; Fan, X.; Bai, J., Enhanced visible light utilization of BiOI/BiOBr/Bi composite catalytic materials for photocatalytic degradation of TC and reduction of Cr (â…¥). Materials Today Sustainability 2024, 27, 100909.
|
Reviewer 4 Report
Comments and Suggestions for Authors
The present study investigates the effects of mannitol dosage and synthesis temperature on the structure, morphology, optoelectronic properties, and photocatalytic performance of BiOI nanosheets (NSs), as well as their photocatalytic mechanism. The manuscript provides interesting and valuable insights for the Nanomaterials journal community; however, several issues need to be addressed before it can be considered for publication:
(1) It is recommended to reorganize the figures in the manuscript. Specifically, since only the BiOI-0.25-60 and BiOI-0-70 samples were subjected to in-depth characterization, the original Fig. 6 should be presented as Fig. 1 and so on. Additionally, the authors should justify why these two samples were selected for detailed characterization. This change would require rewriting portions of the manuscript to ensure the subsequent results and discussion are focused on comparing these two samples.
(2) Any references to Chinese reports throughout the manuscript should be replaced with their English equivalents.
(3) The manuscript does not clearly explain if the BiOI-0.25-60 sample was the only one exhibiting p-type semiconductor behavior. Why was p-type behavior observed specifically under the conditions of 0.25 g of mannitol and a synthesis temperature of 60ºC?
(4) The authors should elucidate the reaction pathway or mechanism through which mannitol induces the p-type behavior in BiOI. This critical aspect is currently missing from the discussion.
(5) The authors claim that the specific surface area of the samples increased in the presence of mannitol (from 12 to 24 m²/g). However, based on the shape of the isotherm (Fig. 2a), the material appears to be non-porous. Please clarify this discrepancy and provide additional evidence to support the claim.
(6) According to the literature (J. Ind. Eng. Chem. 81 (2020) 237–268), BiOI is typically a p-type semiconductor. However, the authors report that the BiOI-0-70 sample exhibits n-type behavior. Why was this transformation observed? Moreover, why was BiOI-0.25-60 the only sample to exhibit p-type behavior, and how do the precise mannitol dosage and synthesis temperature contribute to this transformation?
Author Response
Comments 1: It is recommended to reorganize the figures in the manuscript. Specifically, since only the BiOI-0.25-60 and BiOI-0-70 samples were subjected to in-depth characterization, the original Fig. 6 should be presented as Fig. 1 and so on. Additionally, the authors should justify why these two samples were selected for detailed characterization. This change would require rewriting portions of the manuscript to ensure the subsequent results and discussion are focused on comparing these two samples. |
Response 1: Thank you very much for your valuable comments. We understand your suggestion to rearrange Figure 6 as Figure 1 to better align the order of presentation with the actual sequence of experiments and tests. This would more clearly demonstrate why some tests present data for all BiOI nanosheets, while others compare only BiOI-0.25-60 and BiOI-0.70. In our normal design process, we characterize all BiOI samples with XRD and SEM to confirm that they meet the required structure and morphology. We then perform photocatalytic degradation experiments on all samples. Following this, we select the samples with the best and worst photocatalytic performance, BiOI-0.25-60 nanosheets and BiOI-0.70 nanosheets (which was prepared without mannitol), for detailed analysis of structure, morphology, optoelectronic properties, and even photoelectron transfer mechanisms to explore the factors influencing photocatalytic activity, particularly the impact of mannitol usage on the photocatalyst’s structure, morphology, optoelectronic properties, and photocatalytic performance. However, we wish to clarify that the reason for placing the composite images of XRD, IR, XPS in Figure 1, and the photocatalytic degradation data in Figure 6 is to follow a standard narrative style in scientific reporting. This approach systematically delves deeper into the material’s structure, morphology, optoelectronic properties, photocatalytic performance, and mechanisms, which aligns with readers' usual expectations and reading habits. We hope this explanation clarifies our rationale, and we appreciate your guidance in improving our manuscript.
|
Comments 2: Any references to Chinese reports throughout the manuscript should be replaced with their English equivalents. |
Response 2: Thank you very much for your valuable feedback on the manuscript. I would like to apologize for the inconvenience caused by the appearance of Chinese characters in the references section, which resulted from encoding errors within our submission system. Please rest assured that I have thoroughly reviewed and corrected these issues. I will take extra precautions to ensure this does not recur in the revised submission. Thank you for your understanding and your continued guidance. Incorrectly formatted references have been re-cited and reorganized, especially references 57-66 on lines 185-203.
Comments 3: The manuscript does not clearly explain if the BiOI-0.25-60 sample was the only one exhibiting p-type semiconductor behavior. Why was p-type behavior observed specifically under the conditions of 0.25 g of mannitol and a synthesis temperature of 60ºC? Response 3: Thank you for your comment seeking clarification on the semiconductor behavior of the BiOI-0.25-60 sample. We appreciate the opportunity to clarify that BiOI-0.25-60 is not the only sample exhibiting p-type semiconductor characteristics. In our experiments, all samples prepared with mannitol, regardless of the synthesis temperature or the specific amount of mannitol used, exhibited p-type behavior, as demonstrated in Supplementary Figure S1. The choice of 0.25 g of mannitol and a synthesis temperature of 60°C was specifically highlighted because these conditions were found to be optimal in our study for achieving the desired structural and photocatalytic properties. The presence of mannitol during the synthesis process is critical in determining whether the BiOI nanosheets exhibit p-type or n-type semiconductor characteristics, which is further elaborated in line 361-367 of the revised manuscript. Figure S1 M-S curves of (a) BiOI-0.25-40 and (b) BiOI-0.1-70 Comments 4: The authors should elucidate the reaction pathway or mechanism through which mannitol induces the p-type behavior in BiOI. This critical aspect is currently missing from the discussion. Response 4: Thank you for your valuable feedback regarding our manuscript. We appreciate your suggestion to elucidate the reaction pathway or mechanism through which mannitol induces p-type behavior in BiOI. We recognize the importance of this aspect and agree that it is critical for a comprehensive understanding of our findings. In response to your comment, we have added a detailed discussion on the hypothesized mechanisms by which mannitol influences the electronic structure and energy band arrangement of BiOI, leading to its p-type behavior. This new section aims to provide a clearer understanding of the interaction between mannitol and BiOI at the molecular level. The detailed explanation can be found in line 361-367 of the revised manuscript. We believe this addition will significantly enhance the manuscript and thank you again for helping us improve our work.
Comments 5: The authors claim that the specific surface area of the samples increased in the presence of mannitol (from 12 to 24 m²/g). However, based on the shape of the isotherm (Fig. 2a), the material appears to be non-porous. Please clarify this discrepancy and provide additional evidence to support the claim. Response 5: Thank you for your insightful comments and review of our manuscript. Regarding your question about the apparent discrepancy between the specific surface area increase and the morphology of the isothermal adsorption curves, we appreciate the opportunity to clarify and provide additional supporting evidence. In our study, we observed an increase in the specific surface area of the samples from 12 m²/g to 24 m²/g, which was facilitated by the addition of mannitol. To further explain this phenomenon, we performed BET specific surface area tests and present the data in Figures S2 a-b. Specifically, the pore size of sample n-BiOI-0-70 without added mannitol was 12.827 nm, while the pore size of sample p-BiOI-0.25-60 with added mannitol increased to 23.526 nm. According to the IUPAC classification criteria, both pore sizes fall into the mesoporous range (2-50 nm). Although the isothermal adsorption-desorption curves in Fig. 2a of the revised manuscript do not show obvious mesoporous characteristics, our pore size analysis results still indicate that the pore structure and specific surface area of the materials were indeed significantly improved by the addition of mannitol. We hope that these additional data will answer your questions and further support our conclusions about the changes in the pore properties of the materials. Thank you again for your valuable review and suggestions. Figures S2 BET report for (a) BiOI-0-70 and (b) BiOI-0.25-60
Comments 6: According to the literature (J. Ind. Eng. Chem. 81 (2020) 237–268), BiOI is typically a p-type semiconductor. However, the authors report that the BiOI-0-70 sample exhibits n-type behavior. Why was this transformation observed? Moreover, why was BiOI-0.25-60 the only sample to exhibit p-type behavior, and how do the precise mannitol dosage and synthesis temperature contribute to this transformation? Response 6: Thank you for your insightful question concerning the semiconductor behavior of our BiOI samples. It is well-understood that the semiconductor type of BiOI can be affected by intrinsic defects such as oxygen vacancies or iodine interstitials, which are influenced by the synthesis environment. In our study, the BiOI-0.25-60 sample uniquely exhibited p-type behavior, significantly influenced by the addition of mannitol during synthesis. Mannitol serves as a reducing agent, altering the electronic environment of the material. This control is likely through the reduction of iodine vacancies, essential for sustaining BiOI's p-type characteristics. We would like to clarify that the p-type behavior was not exclusive to the BiOI-0.25-60 sample. All our samples prepared with mannitol demonstrated p-type behavior, irrespective of the synthesis temperature or mannitol dosage, as evidenced in Supplementary Figure S1. The specific conditions—0.25 grams of mannitol at a synthesis temperature of 60°C—were optimized through several experimental iterations. These conditions were found to produce the best results in terms of stability and photocatalytic activity, and they were associated with enhanced p-type behavior. We believe that these optimal conditions help stabilize the crystal structure and promote the formation of beneficial defects that reinforce the p-type properties. This additional information should provide a clearer understanding of how specific synthesis conditions influence the semiconductor behavior observed in our BiOI samples. Thank you again for your detailed review and valuable feedback. |
Round 2
Reviewer 3 Report
Comments and Suggestions for Authors
Accept in present form
Reviewer 4 Report
Comments and Suggestions for Authors
All concerns have been addressed, the necessary revisions have been made, and the manuscript is now ready for publication.